# Development of Immunohistochemical Methods for Casein Detection in Meat Products

**DOI:** 10.3390/foods10010028

**Published:** 2020-12-24

**Authors:** Ludmila Kalčáková, Matej Pospiech, Bohuslava Tremlová, Zdeňka Javůrková, Irina Chernukha

**Affiliations:** 1Department of Plant Origin Foodstuffs Hygiene and Technology, Faculty of Veterinary Hygiene and Ecology, University of Veterinary and Pharmaceutical Sciences Brno, 612 24 Brno, Czech Republic; lida.anezka.l@gmail.com (L.K.); mpospiech@vfu.cz (M.P.); tremlovab@vfu.cz (B.T.); 2The V.M. Gorbatov Federal Research Center for Food Systems of Russian Academy of Sciences, Talalikhina ul. 26, 109316 Moscow, Russia; imcher@vniimp.ru

**Keywords:** milk, immunoreactivity, sausages, allergy, ELISA, food microscopy

## Abstract

To increase production efficiency of meat products, milk protein additives are often used. Despite a number of advantages, use of dairy ingredients involves a certain risk, namely the allergenic potential of milk proteins. A number of methods have been developed to detect milk-origin raw materials in foodstuffs, including immunological reference methods. This study presents newly developed immunohistochemical (IHC) methods for casein detection in meat products. Casein was successfully detected directly in meat products where sensitivity was determined at 1.21 and specificity at 0.28. The results obtained from the IHC were compared with the Enzyme-Linked Immuno Sorbent Assay (ELISA) and there was no statistically significant difference between the IHC and ELISA methods (*p* > 0.05). The correspondence between the methods was 72% in total. The highest correspondence was reached in frankfurters (90%), the lowest in canned pâté (44%).

## 1. Introduction

Nowadays, meat products represent foodstuffs with a wide range of non-meat ingredients. They are used for various reasons, including the economic, technological, or marketing side of production as well as to enhance taste and flavor. One of the non-meat ingredients used in meat products is milk protein. Milk proteins are obtained from milk and dairy products [1]. Within the meat industry, a significant position among the various ground meat products is occupied by sausages, where milk protein additives are utilized as emulsifiers or production cost reducers. Another effect, however, is to improve the yield, slicing quality, flavor, and nutritional value of the final product [2]. Caseins are made up of several proteins (αs1-, αs2-, β-, and κ-) capable of forming micelles (130–160 nm circular structures). The emulsifying properties are explained by forming structures with high surface activity. In the meat industry, caseins are used in variously modified forms. The most widespread form is sodium caseinate that is characterized by its high solubility and ability to increase volume of the meat dough with a higher capacity to bind water. It also has a significant effect as a fat emulsifier during the treatment [3]. Sodium caseinate is often used as an additive for dry-fermented sausages due to its emulsifying properties [4]. Because of its excellent functional properties, addition of milk protein ingredients into ground meat products has a great potential. Examples of successful use in such meat products include caseinate in chicken nuggets, whey protein concentrate in sausages, sodium caseinate with whey powder in Turkish meat rolls, or milk co-precipitates in chicken loaf [5]. Whey proteins are also composed of a number of proteins. Mainly β-lactoglobulin, α-lactalbumin, serum protein, immunoglobulins, and polypeptides. β-lactoglobulin is a globular dimeric protein that represents 50% of total whey proteins. After partial denaturation, it is able to form dimers with other milk proteins. Other functional whey proteins include immunoglobulins, which only account for 2% in whey but have significant emulsifying properties. The most common forms used in meat production are whey concentrates and isolates. An option is likewise the textured whey protein, which is used as a partial replacement of muscle in meat products up to 40% *w*/*w* [3]. Milk-origin raw materials are also used to maintain the juiciness of low-fat meat products. For example, milk powder, sodium caseinate, milk co-precipitate, and skim milk protein are considered to be water-binding substitutes for fat [6].

From the legislative point of view, dairy proteins are considered to be a component of foodstuffs and can therefore replace some additives. Thus, they are included among the essential ingredients in the ingredients list and restrictions on additives do not apply to them. However, according to E.U. legislation, milk is also classified as an allergen. Therefore, its presence must be declared according to the legislation [7].

The incidence of food allergies is estimated to range from 0.5% to 3% within Europe [8]. Nowadays, 0.6–2.5% of preschool children, 0.3% of older children and adolescents, and less than 0.5% of adults suffer from cow’s milk allergy globally. Unfortunately, the prevalence of allergy to cow’s milk is steadily increasing [9].

Cow’s milk contains approximately 30 potentially allergenic proteins. Out of this, about 30 to 35 g are contained in 1 dm3 of milk [10]. The primary cow’s milk allergens include caseins, β-lactoglobulin, and α-lactalbumin. The casein fractions are represented in the form of an isoform of different proportions. These are isoforms such as α-S1-casein (32%), α-S2-casein (10%), β-casein (28%), and κ-casein (10%). Whey proteins include α-lactalbumin, β-lactoglobulin, immunoglobulins, bovine serum albumin, and trace amounts of lactoferrin [8]. It is estimated that 66% of milk allergies are caused by β-lactoglobulin, 57% by casein, and significantly less by α-lactalbumin and bovine serum albumin (18%) [10]. D’Auria et al. [8] report that allergenic activity was observed for β-casein in 35% of patients, for α-S1-casein in 26%, as well as for κ-casein in 26%, whereas for β-lactoglobulin in 19%, for α-lactalbumin in 12%, for lactoferrin in 3%, and for bovine serum in 1%. As with all food allergens, there is currently no suitable treatment for cow’s milk allergy. The only therapy available is to avoid consuming milk and dairy products. However, allergenic proteins are not the only factor responsible for allergic reactivity as well as test immunoreactivity. The food processing, food matrix effect, and different degree of digestion can change the immunoreactivity. Several recent studies of immunoreactivity changes are described by Villa et al. [11]. None of them, nevertheless, describes immunoreactivity changes in meat products where strong matrix effect is expected. Likewise, they do not report effects of thermal processing on a milk-containing food matrix where detection of milk additives can be reduced [12].

The aim of this work was to develop an immunohistochemical method for direct detection of casein in meat products’ matrix and to validate it. Secondary objectives involved verification of immunoreactivity of common types of milk additives used in meat industry and demonstrating the masking effect of different meat matrices on the immunoreactivity of immunoassay applied.

## 2. Materials and Methods

Common milk additives available in the market were used. Specifically, calcium caseinate, whey protein with high lactose content, thermostable whey concentrate, and whey concentrate containing 85% of casein (MEGGLE Wasserburg GmbH & CO.KG, Wasserburg, Germany), as well as milk powder, whey powder, and buttermilk powder (LAKTOS, a.s., Prague, Czech Republic).

Two groups of model samples were prepared. The meat bases of the model samples were prepared from minced pork and beef meat in a 1:1 ratio. The first group was prepared with the addition of different types of milk additives in the concentration of 2.5%, which is a common addition into meat products to reach the required effect [3]. The additives used were calcium caseinate (C), whey powder (W1), whey protein with high lactose content (W2), thermostable whey concentrate (W3), buttermilk powder (W4), whey concentrate containing 85% of casein (W5), and milk powder (M1). The second group (samples C1–C7) was prepared with increasing concentrations of calcium caseinate (0.0001%; 0.001%; 0.01%; 0.1%; 1%; 2.5%; and 10%), and it was used to determine the limit of detection according to previous study [13]. The raw materials were ground and mixed in Vorwerk Thermomix 31 (Vorwerk, Wuppertal, Germany) at 7000 rpm for 2 min. Subsequently, each meat matrix was quantitatively transferred to a sample container and cooked in water bath until the core matrix was cooked at 70 °C for 10 min (Rommelsbacher, Dinkelsbühl, Germany).

For immunohistochemical methods, the samples were prepared according to the following procedure: four partial samples were frozen at −35 °C for each sample. After freezing, the samples were cut into 10 μm sections on a rotation microtome HM 550 (Microm, Walldorf, Germany). The partial samples were cut into two serial sections, while 50 µm were overcut. Totally eight sections for each sample were prepared. The sections were mounted on SuperFrost plus slides (Menzel-Glasser, Menzel GmbH & Co KG, Braunschweig, Germany). Before the immunohistochemical procedure, the slides were dried in a fridge. The immunohistochemical procedure followed in no later than seven days.

The immunohistochemical procedure was based on the IHC method described by Pospiech, Tremlová, Renčová, and Randulová [13] and Řezáčová-Lukášková et al. [14] with the following changes. The immunohistochemical VECTASTAIN Elite ABC KIT was used (VectorLaboratories, PK 6101, Burlingame, CA, USA). Rabbit anti-beta casein polyclonal antibody (Bioss Antibodies, Woburn, MA, USA) was used as the primary antibody, and antibody diluent (DakoCytomation Ref.S0809, Glostrup, Denmark) was used for dilution. In negative control, the primary antibody was substituted by antibody diluent (DakoCytomation ref. S0809). For antigen retrieval, Citrate-EDTA was used (Sigma-Aldrich, St. Louis, MO, USA) according to Bednářová, Pospiech, Tremlová, Řezáčová-Lukášková, and Bednář [15]. Avidin-biotin complex was used for multiplication of signal, the second phase included biotin-conjugated secondary antibody (VectorLaboratories, PK 6101, Burlingame, CA, USA), and the third phase applied avidin-biotin complex with horseradish peroxidase (VectorLaboratories, PK 6101, Burlingame, CA, USA). For immunological visualization, peroxidase substrate HistoGreen (Linaris GmbH, Mannheim, Denmark) and 3,3′-diaminobenzidine (DAB) (DakoCytomation, Glostrup, Denmark) were used. For background staining, Calleja, Toluidine Blue, and Nuclei Fast Red (MERCI s.r.o., Brno, Czech Republic) were utilized.

To analyze repeatability and reproducibility, two model samples were examined. The first guaranteed a positive (1% addition of 94% calcium caseinate) (R1) and the second guaranteed a negative pattern (without any milk content additive (R2). To verify repeatability, both analyzed samples (samples R1 and R2) were repeatedly examined five times on different days. To verify reproducibility, samples of R1 and R2 were examined also by another trained examiner. To analyze specificity, 25 meat products from the Czech and Russian market with and without declared content of milk addition were analyzed twice by two trained examiners. The meat products were chosen randomly in the retail market and comprised frankfurters (10), canned pâté (9), cooked salami (2), meat loaf (2), and ham mousse (2). Frankfurters M2–M7 were prepared by Russian normative technology according to COST P 52196-2011. These products contained 3% of dry cow defatted milk. The Czech producer confirmed milk addition in the analyzed products without quantitative declaration. The products quantity represents the frequency of milk or milk protein usage in the product groups. All validation criteria were evaluated after sample anonymization. According to laboratory practice, a sample is evaluated as positive in case of chromogen precipitation in minimally three histological sections where each IHC examination evaluated eight histological sections.

IHC results were confirmed by sandwich ELISA kit of Veratox^®^ For Casein Allergen Test 8460 (Neogen, Lansing, MI, USA). The samples from the market were examined according to the kit procedure. Before weighting a 5 g portion, 100 g of the samples were homogenized.

Statistical analyses were performed in Unistat 6.1. (Unistat Ltd., London, UK) and MS Excel (Microsoft Corporation, Washington, DC, USA) software. To determine the significance differences of the IHC and ELISA, McNemar’s test [16] utilizing test criterion calculation of χ^2^ (chi quadrate) was used.

## 3. Results and Discussion

### 3.1. Method Development

In immunohistochemical methods, there are several critical steps that can negatively affect the outcome of the immunohistochemical reaction. When developing a new method, it is therefore necessary to verify these critical steps before validating and verifying the method itself. The key step is not just the selection of a suitable primary antibody but also its concentration. A polyclonal primary antibody was chosen because this type is commonly used in most commercial kits and it is readily available. The recommended dilution for IHC application was determined by the primary antibody manufacturer at 1:200 to 1:400, for ELISA and western blotting at 1:300 to 1:1000 dilution. For immunohistochemical methods, higher concentrations are used due to lower sensitivity compared with ELISA or western blotting. Within the method development, dilution of the primary antibody was verified at 1:10; 1:100; 1:500; 1:1000; and 1:1500. The result is presented in Table 1.

The results were evaluated qualitatively, the optimal dilution proved to be the 1:500 dilution. With regard to economic considerations, dilutions of 1:1000 and 1:1500 may also be used, but only in the case of expected high milk protein content. If milk proteins are used as functional additives in the meat product, concentrations of up to 2% of caseinates and up to 3.5% of whey proteins can be expected [17], thus a dilution of 1:1000 would be sufficient. However, in the case of cross-contamination monitoring, a 1:500 dilution is strongly recommended with regards to the low concentrations.

For proper interpretation of the signal in IHC of immunohistochemical methods, a sufficient contrast between the observed signal (antigen) and other structures (background) is also critical. Caseinate forms a gel-like structure overlapping the muscle proteins [18]. Thus, it is important to find sufficiently contrasting targeted background stainings over the selected chromogen with regard to the fact of color bonding where caseinate overlaps the muscle protein. In order to obtain the strongest signal possible, two chromogens and three different stainings were compared in the C6 sample. The results are presented in Table 2.

As described by Thomas and Lemmer [19], DAB and HistoGreen chromogens have the same sensitivity. However, when comparing these two with peroxidase substrates, differences in contrast are evident. DAB provides a clear and smooth but sharply bounded and consistent form of immunoreactive structures, and it is typically characterized by regular incision rates in terms of intensity and density. HistoGreen creates coarse grain structures and tends to change the color quickly. The choice of DAB chromogen with background Toluidine Blue staining was based on the work by Řezáčová-Lukášková et al. [14], where high contrast was achieved for wheat protein. Likewise, high contrast was shown using the DAB chromogen with Calleja staining [13] for soy protein. These assumptions were not confirmed as part of the dairy protein detection. DAB chromogen highlighted the presence of milk proteins in brown. However, in combination with Calleja staining, they did not achieve sufficient contrast (Figure 1), as the other structures of the sample (especially the muscle) are colored yellow-green, from green to brown.

The combination with Toluidine Blue achieved a different color spectrum of the background (Figure 2).

The Toluidine Blue stains other structures blue, blue-violet to blue-gray. Because none of the stainings achieved sufficient contrast, Fast Nuclei Red staining was tested as a new staining for IHC. Using Fast Nuclei Red in combination with DAB chromogen did not achieve sufficient contrast. Sufficient contrast was achieved only with the Fast Nuclei Red in combination with HistoGreen chromogen (Figure 3).

An alternative to immunochemical methods includes immunofluorescence methods, which do not need background staining, as a specific wavelength emission is measured [20]. However, thereby we lose accompanying information about the protein formation and, in particular, localization of the monitored protein in the product structure.

### 3.2. Method Validation

The qualitative methods (i.e., methods determining the presence or absence of the analyte/allergenic compound) are common for laboratory examination, especially as screening methods prior to a more expensive examination by a quantitative method. The advantages of their use are the costs-reduction and time-saving [21]. Similarly, in an official document by the European Commission [22] and Magnusson [23], the following parameters were provided: limit of detection, repeatability and reproducibility, sensitivity, and selectivity/specificity.

### 3.3. Limit of Detection (LOD)

Determination of the milk proteins added to meat products is not studied very much. Studies based on immunological methods most often deal with the detection of adulteration of different types of milk or cheeses with cow’s milk. The detection limit of these methods ranges from 0.001% (*v*/*v*) to 15% (*v*/*v*) [24,25] and depends on the method used and the antigen chosen using different milk proteins or immunoglobulins. Commercial immunoassay kits, where the detection limit ranges from 0.1–5 mg kg^−1^ are currently available. Likewise, the limit of detection (LOD) is dependent on the method used, the antigen selected, and also on the analyzed matrix [26]. LOD in our developed IHC was determined at 0.0001% (*w*/*w*) of caseinate that represents 9.4 mg kg^−1^ of casein (Table 3).

According to Trullols, Ruisanchez, and Rius [21], LOD was determined as positively evaluated samples in all eight replicates. The detection limit of the applied ELISA method was determined at 2.5 mg kg^−1^ of casein according to its manufacturer. Analytical methods, such as ELISA, are used routinely to obtain semiquantitative or quantitative results. For the determination of milk, sandwich ELISA kits with sensitivity of up to 1 mg kg^−1^ are commercially available. There are also kits that are formed only to detect a specific milk protein (bovine serum albumin, casein, beta-lactoglobulin) and others to detect whole milk or whey protein with the achievement of LOD values within the range of 0.5 to 5 mg kg^−1^ [27]. To quantify casein in food products (e.g., cured sausages), Spizzirri and Crillo [28] used the S-ELISA (sandwich ELISA) method using rabbit and goat anti-casein antibodies where LOD was 0.5 mg kg^−1^. According to Gern, Yang, Evrard, and Sampson [29], the presence of 136 μg mL^−1^ of casein and 5 μg mL^−1^ of whey was enough to induce allergic reactions. It means that the developed and commonly used methods are adequate for milk and whey protein detection in case of consumer protection.

### 3.4. Repeatability and Reproducibility

Repeatability and reproducibility are optional parameters for qualitative methods. In case of IHC assays, where the protocols are performed on different days, the impact of preanalytical factors were described in many studies [30,31]. The repeatability and reproducibility were provided for the evaluation of the environmental influence. The repeatability was established to be 100% for positive as well as negative samples (Table 4). Based on personal factors during examination, the reproducibility was performed. The reproducibility was also established as 100% (Table 5).

### 3.5. Sensitivity and Specificity

Fuchs [32] reports that allergy to cow’s milk proteins is probably the most common food allergy affecting the European population. Milk allergy is, however, caused by several proteins that differ not only in their allergenic potential but also in post-processing stability. Whey proteins of β-lactoglobulin as well as serum albumin and immunoglobulin lose their allergenic potential during heat treatment and enzymatic cleavage. In contrast, caseins and α-lactalbumin are considered thermostable. This applies mainly to casein with a loose tertiary, highly hydrated structure. Caseins are not significantly affected by severe heat treatment but are very susceptible to all proteinases and exopeptidases [33]. This fact is also confirmed by Spizzirri and Crillo [28] claiming that casein is a thermostable protein; in contrast, whey β-lactoglobulin is thermally labile and irreversibly denatured or aggregated with casein micelles and α-lactalbumin after heat treatment. However, the research by Bu, Luo, Zheng, and Zheng [34] disproves the claim that thermosetting proteins are thermolabile. In their study, they found that α-lactalbumin and β-lactoglobulin have even higher IgG binding after pasteurization and sterilization than before heat treatment. β-casein is able to maintain its immunoreactivity even after warming up to 55 °C [9]. For meat products, cooking at 70 °C for 10 min is mostly used. Nevertheless, based on a number of studies, we can assume that even such a warm-up will not cause a sufficient reduction in the allergenicity of caseins. Different immunoreactivity was also confirmed for the analytical methods. In ELISA, extraction effectivity is important [35] in comparison with IHC using direct detection of antigen in food matrix where extraction effectivity is negligible for solid antigen. The solubility as well as immunoreactivity can be changed by protein aggregation. The aggregate can be detected by the IHC method easily, as it is known from histological diagnostic as well as in cases of decreasing immunoreactivity. The important issues for immunoreactivity of IHC methods are pre- and postanalytical factors [31]. For confirmation of immunoreactivity, various dairy additives were tested. Immunoreactivity was evaluated on model samples with 2.5% content of additives: calcium caseinate (C1–7), whey protein (W1–W5), and milk addition (M1). For all additions, the casein immunoreactivity was confirmed (Table 6).

This result points to the suitability of polyclonal antibodies for the IHC and generally supports immunological based methods. The content of casein in dry whey protein concentrate was confirmed also in another study [36], and it is in conformity with our results. Heat treatment, however, is not the only way that can affect the immunoreactivity of raw materials used. In the production of dried sausages, there is a number of reactions that make sausages obtain their characteristic texture, flavor, and smell. They include in particular hydrolysis and autooxidation of lipids, proteolysis, and transformation of amino acids to aromatic compounds. In addition, the fermentation, such as lactic acid bacteria, has a major influence on the formation of small peptides and free amino acids. Dried sausages often include sodium caseinate, which undergoes transformations similar to meat protein (i.e., the formation of small peptides and free amino acids). These small peptides contribute to the development of the characteristic taste of dry fermented products and also fulfill important bioactive functions, such as antioxidant and antihypertensive activity in dried meat products. Within analysis of casein degradation, it was found that most of the peptide residues in the food remain from β-casein [4]. Many studies also focus on the values of the minimum observed dose inducing allergic reactions to milk and its products. Several of these studies have demonstrated severe reactions to very small amounts of these allergens. An example is a severe anaphylactic reaction that was caused by consumption of a sausage containing cow’s milk (corresponding to casein content of 60 mg) [27]. In that case, casein was selected as the most suitable protein for IHC detection. Considering casein sensitivity to proteinases, an enzymatic method cannot be used as one of the antigen retrieval techniques [15]. It is not easy to verify the various effects that can occur in foodstuffs. Demonstration of truly positive and negative samples is commonly used for this purpose. For the purpose of validation in a diverse matrix of meat products, products from the real retail market were evaluated according to Trullols, Ruisanchez, and Rius [21]. The results are presented in Table 7. Correspondence level of the IHC method with ELISA for all the analyzed meat groups reached 72% in total.

The IHC sensitivity was set at 1.21 and the specificity at 0.28 according to Trullols, Ruisanchez, and Rius [21]. Sensitivity is the ability of a method to detect truly positive samples as positive. It can be calculated as test positives/total number of known positives. Specificity is defined as the ability of a method to detect truly negative samples as negative. It can be expressed as test negatives/total number of known negatives [Trullols et al., 2004]. Model meat products with various additive concentrations, different dairy additive types, and products from the retail network were used for the calculation. The method is therefore highly sensitive for the detection of milk additives. The concentration does not show a strong impact on the developed methods. In contrast to the IHC method, the ELISA kit reaches sensitivity of 1.15 and specificity of 0.4. The specificity in both methods should be considered as indicative because only two negative samples were evaluated. Thus, in contrast to the ELISA method, the IHC method is less sensitive.

McNemar’s Test (Chi-square statistic) was used to compare the qualitative results of both methods applied [13,21] where model samples and meat products were evaluated. The comparison shows that even with respect to certain differences, especially in the number of negatively evaluated results by IHC (*n* = 7) and ELISA (*n* = 5), including dubious samples, both methods are comparable (*p* > 0.05) (Table 7 and Table 8).

The reason for choosing ELISA for the comparison was that it is commonly used for the determination of milk proteins in foodstuffs and it is also the official screening method in many countries [37]. For meat products, determination of defatted milk powder was also validated already at 10 mg kg^−1^ [38]. These authors also reported a relatively high level of reproducibility of interlaboratory accuracy (Allergeneye ELISA) for milk with the relative standard deviation for reproducibility (RSDR) amounting to 6.8–10.5% for all processed foodstuffs. Similar results in interlaboratory reproducibility (RSDR 7–14% and 12–17%) were achieved by Matsuda et al. [37], who tested the presence of milk in sausages using ELISA methods: FASTKIT and ELISA FASPEK. However, compared with the above, the ELISA method is still not considered to be sufficiently perfect. A perfect method is expected to incorporate features such as high accuracy and sensitivity, low costs, and rapid analysis of various types of technologically processed food products. There is a variety of commercially available ELISA kits; nevertheless, none of them meets the above requirements. The biggest problem is false-negative results in heat-treated products [39]. The false-negative results were demonstrated also in this work where ELISA reached 15.3% and IHC 19.2% in total. In frankfurters, ELISA reached 10% and IHC 0%. In canned pâté, ELISA reached 33.3% and IHC 55.5% of false-negative results.

Lactose was declared in a chorizo product (L1) where it is added as part of the starter to improve the fermentation process using lactic acid bacteria [40]. Our results obtained by ELISA and IHC were positive, contrary to the original assumption. The positivity of the sample with declared lactose content can be caused by contamination of this additive during the production. Contamination of the lactose additive by milk proteins might be due to its production from cow’s milk or whey. Lactose is used not only in the food industry but also in the pharmaceutical industry where it is used as an ineffective ingredient, a filler in many pharmaceutical preparations. The problem of lactose contamination by milk proteins has already been recognized in the past in pharmaceutical products [41]. Therefore, five different whey products commonly used in the food industry were tested—Table 5 (W1–W5). Both methods, the IHC as well as ELISA, confirmed the presence of casein in these products. Our study confirmed also the presence of casein in lactose additive. A small content of casein in lactose was also confirmed not only among various manufacturers but also among individual batch preparations from the same manufacturer. From this point of view, we can assume that the sample from the market network examined by us was impacted by a casein contamination of lactose or unsatisfactory purification during lactose producing.

Canned pâté products also differed from the assumptions, where the IHC method did not confirm presence of milk additive. This included a doubtful sample of M12 and false-negative samples of M11 and M16–M18, which represents 44%. In this group of products, milk additives are primarily used as emulsifiers [3,4]. This property and high degree of homogenization of the matrix are attributed to the negative result of the IHC assay, when milk proteins are surrounded by the fat particles and subsequently form an emulsion matrix with water and fat [42]. This phenomenon leads to the spreading of proteins on a larger area, and the resulting precipitate is of weaker intensity. With respect to the non-specific binding always present, the signal intensity is then lost. A similar result was also shown by the ELISA examination where M13, M15, and M17 were false-negative (33%). Our results are in agreement with Zaffran and Sathe [43], who refer that immunoreactivity can be changed by food processing. Weakening of signal intensity could also be due to denaturation of proteins that might be caused by higher temperatures and pressure used in the production of canned meat products. The food processing with protein spreading also can explain false negative results of IHC. This hypothesis, however, is in contradiction to many authors who have demonstrated the allergenic potential of caseins in heat-treated milk even after temperatures above 170 °C were applied for up to 60 min [44]. As reported by D’Auria et al. [8], caseins are resistant to high temperatures, and it can be said that they represent most of the proteins contained in heat-treated foodstuffs. Heat resistance can be affected by thiol group binding κ-casein [45]. However, low immunoreactivity of canned pâté remains to be further explored. Dupont et al. [44] describe the possibility of reducing allergenicity and allergic reactions of children consuming heat-treated milk. When evaluating the allergenicity, the team of authors came to the finding that caseins appeared to be heavily allergenic even at temperatures of 170 °C for up to 60 min of heating. Although α-casein is thermally stable, it is influenced by the presence of phosphate, which is essential for the proteins’ function of binding calcium. Because caseins are resistant to high temperatures, they can be said to be the major allergenic proteins contained in heat-treated foodstuffs [8]. This statement was also confirmed in other heat-treated meat products (frankfurter type) analyzed in our study. Comparison of both methods shows that the IHC method is comparable with ELISA for this group of meat products. Of the 10 analyzed samples of frankfurter-type products, correspondence was achieved in 9 samples (90%) because ELISA showed a false-negative response in one case. On the other hand, canned pâté correspondence reached 40%. The correspondence for ham mousse was 100%, for salami 50%, and for meat loaf 100% (Table 7). However, in these three product groups, only a small number of samples was evaluated, and data are only approximated. Further studies should focus on immunoreactivity changes in pâté and other processed meat products.

## 4. Conclusions

Milk proteins are a frequent additive especially to emulsified meat products. The immunohistochemical method for the detection of casein was developed and validated in this work. The immunohistochemical methods using the HistoGreen chromogen and Nuclear Fast Red background staining were evaluated as the most appropriate. The detection limit of the method was determined at 9.4 mg kg^−1^, repeatability and reproducibility were established at 100%, sensitivity was 1.21, and specificity was 0.28.

The casein, as one of the stable milk proteins, was confirmed as an appropriate antigen for immunologically based tests. IHC as well as ELISA detected casein in calcium caseinate, whey powder, whey protein with high lactose content, thermostable whey concentrate, buttermilk powder, whey concentrate containing 85% of casein, as well as milk.

Nevertheless, the impact of food processing to casein immunoreactivity was observed. The canned pâté demonstrated the highest masking effect of its matrix for booth methods. ELISA reached 33.3% and IHC 55.5% of false-negative results. The lowest masking effect was confirmed for whole meat products following meat loaf, salami, and frankfurters. This study is a pilot study for evaluation of food processing impact on immunoreactivity using the direct antigen detection in food matrix.

## Figures and Tables

**Figure 1 foods-10-00028-f001:**
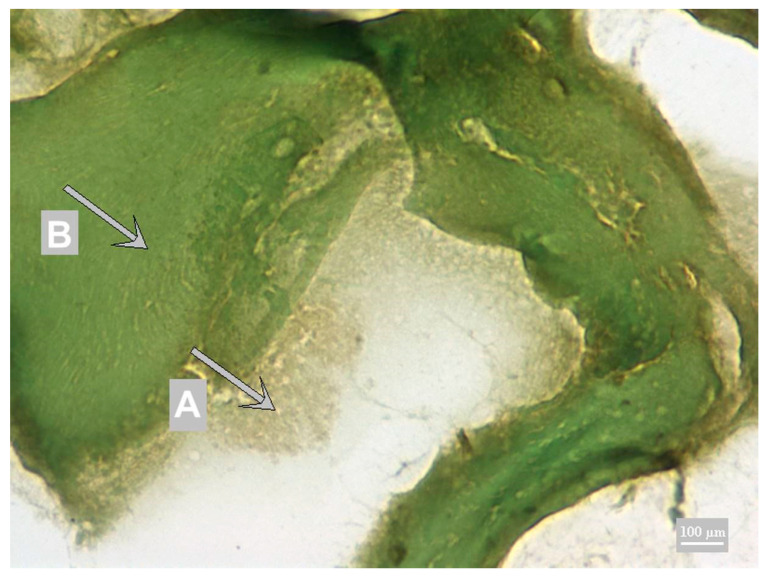
Calleja staining with DAB chromogen; A chromogen precipitate—calcium caseinate (gray-brown); B muscle fiber (green).

**Figure 2 foods-10-00028-f002:**
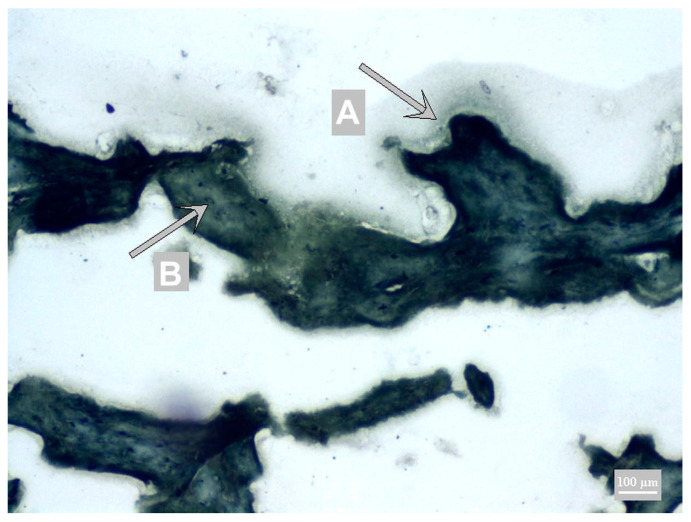
Toluidine Blue with DAB chromogen; A chromogen precipitate—calcium caseinate (blue-brown); B muscle fiber (blue).

**Figure 3 foods-10-00028-f003:**
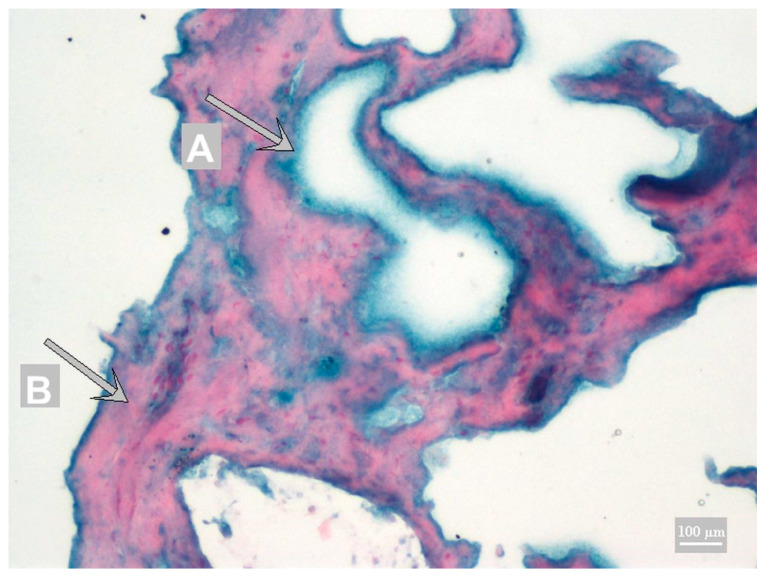
Fast Nuclei Red with HistoGreen chromogen; A chromogen precipitate—calcium caseinate (green); B muscle fiber (red).

**Table 1 foods-10-00028-t001:** Effect of primary antibody dilution on IHC detection of beta-casein in selected model samples.

Model Sample	Caseinate Addition	Primary Antibody Dilution				
		1:10	1:100	1:500	1:1000	1:1500
**C5**	1%	+++	+++	+++	+++	+++
**C6**	2.5%	+++	+++	+++	++	+

+: weak signal; ++: strong signal; +++: very strong signal; C5 and C6: calcium caseinate.

**Table 2 foods-10-00028-t002:** Contrast between chromogens and background staining in calcium caseinate 2.5% model sample.

Chromogens	Background Staining		
	Calleja	Toluidine Blue	Fast Nuclei Red
**DAB**	+	+	-
**HistoGreen**	-	-	+++

+: weak signal; +++: very strong signal.

**Table 3 foods-10-00028-t003:** Limit of detection.

Caseinate Addition (%)	Quantity of Casein (mg kg^−1^)	No. of Repetitions	Positive/Negative
0	0	8	0/8
0.0001	9.4	8	8/0
0.001	94	8	8/0
0.01	940	8	8/0
0.1	9400	8	8/0
1	94,000	8	8/0
10	940,000	8	8/0

**Table 4 foods-10-00028-t004:** Repeatability of IHC examination.

Repetition	R1	R2
1	Yes	No
2	Yes	No
3	Yes	No
4	Yes	No
5	Yes	No
Correspondence	100%	100%

R1: positive control; R2: negative control.

**Table 5 foods-10-00028-t005:** Reproducibility of IHC examination by different examiners.

Examiner	R1	R2
A	Yes	No
A	Yes	No
A	Yes	No
A	Yes	No
A	Yes	No
B	Yes	No
B	Yes	No
B	Yes	No
B	Yes	No
B	Yes	No
Correspondence	100%	100%

A: examiner 1; B: examiner 2; R1: positive control; R2: negative control.

**Table 6 foods-10-00028-t006:** IHC and ELISA immunoreactivity to different additive types in model meat products.

Sample	C1	C2	C3	C4	C5	C6	C7	W1	W2	W3	W4	W5	M1
**IHC**	+	+	+	+	+	+	+	+	+	+	+	+	+
**ELISA**	+	+	+	+	+	+	+	+	+	+	+	+	+

Note: C1–7: calcium caseinate; W1: whey powder; W2: whey protein with high lactose content; W3: thermostable whey concentrate; W4: buttermilk powder; W5: whey concentrate containing 85% of casein; M1: milk; +: positive reaction.

**Table 7 foods-10-00028-t007:** IHC and ELISA immunoreactivity to casein in commercial meat products.

Meat Product																										
Type of Additives and Identification		M2	M3	M4	M5	M6	M7	M8	M9	M10	M11	M12	M13	M14	M15	M16	M17	M18	M19	M20	Co1	Co2	Ch1	Ch2	Ch3	L1
IHC		+	+	+	+	+	+	+	+	+	− *	+/− *	+	+	+	− *	−	− *	+	+	−	−	+	+	+	+
ELISA		+	+	+	+	− *	+	+	+	+	+	+	− *	+	− *	+	−	+	+	+	−	+ **	+	+	+	+
Meat Product		FR						CP	HM		CP								FR			SA	ML	FR	ML	SA
**Correspondence of IHC and ELISA methods**																										
Meat Product	Total (**%**)																									
FR	90	5/6 (83%)																	3/3 (100%)					1/1 (100%)		
CP	44							1/1 (100%)			3/8 (37.5%)															
HM	100								2/2 (100%)																	
SA	50																					0/1 (0%)				1/1 (100%)
ML	100																						1/1 (100%)		1/1 (100%)	
All Types	72										18/25 (72)															

M2–20: milk; Co1–2: control sample without any milk protein; Ch1–3: cheese; L1: lactose; +: positive; −: negative, +/−: dubious reaction; *: false negative; **: false positive; FR: frankfurters; CP: canned pâté; HM: ham mousse; SA: salami; ML: meat loaf; IHC/ELISA (% correspondence).

**Table 8 foods-10-00028-t008:** IHC and ELISA comparison by Chi-square statistic.

	IHC+	IHC−	Sum
ELISA+	29	5	34
ELISA−	3	2	5
Sum	33	7	40
*Chst*	0.5		
*p* =	0.4795		

Dubious samples were calculated as positive and also negative ones.

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
