# Peer review of "Development of Immunohistochemical Methods for Casein Detection in Meat Products"

_foods, 2020, doi:10.3390/foods10010028_

Round 1

Reviewer 1 Report

The present paper involves the development of a Immunohistochemical (IHC) method to detect the presence of caseins in meat products.

As related in the conclusion section, this method seems to be an initial part of a research that involves the use of this method to evaluate changes in casein immunoreactivity. Many other publications have deal with the development of methods to detect milk proteins in processed foods, so novelty is compromised. Consequently, authors should revise the real objective in the introduction, and also, state the real application beyond further research purposes, and answer: Why this method is different of better than the already available ones?

Related to this, currently, fast and reliable qualitative methods, as screening methods prior to examination by quantitative methods (as authors intend for the developed one in lines 224-226) are already available in the form of Immunochromatography or IEF strips, with a proper sensitivity (1-2 ppm), much cheaper, easy (not equipment required) and fast (10 min) than the related method. So, this method does not represent an improvement over other available methods. In my opinion, according to the expected use, comparison should have been performed with these methods (qualitative) and not with ELISA (commonly quantitative).

Anyway, a proper comparison with commercial ELISA has been performed. However, as related in the results and discussion section, ELISA has better sensitivity and specificity, so why don’t use this kind of methods for the evaluation of changes in immunoreactivity? This should be taken into account in the discussion section. Finally, if visualization of the casein in the meat matrix is the only factual advantage, this improvement cannot be considered as enough “novelty”, as concluded before.  

Consequently, major changes must be performed in the manuscript, in order to highlight the novelty, advantages and usefulness of the proposed method.

Also, these minor comments should be considered:

In the introduction, authors call caseins as “milk proteins” (line 31). This is confusing. The two groups of milk proteins are caseins and whey proteins. Milk proteins are all of them.

For References 1 to 10, please, revise the numbers of references in the text, that must be situated between square brackets, but in many cases, these brackets are lost (e.g. lines 37, 40, 47, 48… introduction in general, and 282, 308…).

Author Response

Dear reviewer,

Thank you for these pertinent comments

We agree that aim and conclusions were not specified in details. We revised the major objective in the introduction, and we defined two secondary objectives. The first objective was verification of immunoreactivity of common types of milk additives used in meat industry and the second to demonstrate the masking effect of different meat matrices on the immunoreactivity of immunoassay applied. L101-L105. According to these changes the conclusion was also revised L429-L448.

The importance of this investigation was not only developing the new direct methods but also its importance to show how meat products can change immunoreactivity of immunoassay. We changed accordingly the introduction and discussion L87-L95, L301-L305; L344-350. We understand that the developed method is not better than ELISA, but statistical evaluation showed the developed method to provide similar performance and possibility to be used for direct observation of antigen in the analysed samples. 

The reason for choosing ELISA for the comparison was that it is commonly used for the determination of milk proteins in foodstuff and it is also the official screening method in many countries. Used methods were interpreted qualitatively to demonstrate more relevant comparison because according to this kit qualitative as well quantitative interpretation is possible.

The minor comments

1) To make the introduction more consise, as requested by other reviewer, the confusing sentences were deleted L30-L37.

2) The citation format was corrected. We are sorry for that mistake.

We also performed professional editing to ensure that the text is well phrased and free from typographical and grammatical errors. All changes are highlighted in the manuscript.

Reviewer 2 Report

This manuscript titled "Development of Immunohistochemical Methods for Casein Detection in Meat Products" developed immunohistochemical (IHC) methods for casein detection in meat products. The results found that the casein could be detected directly in meat products, which is reaching a similar level compared with the ELISA test. This topic is very interesting and can provide some potential detecting methods for the food industry. However, some questions need to be addressed. 

  1. The introduction section is too long, please simplify this section and describe your research goals clearly. 
  2. For the citation format in the introduction section, please replace "1, 2, 3, 4...." with "[1], [2], [3].....". Line 33-35, please add at least one reference to support your statement.
  3.  Line 105-115, why did the authors prepared samples in those ratios and concentrations? Please cite the relevant citations in this section.
  4. Line 408-425: Please simplify this section and just describe the major findings obtained from this work.  If it's possible, please add some information regarding your future further research.
  5. References: Please double check the format of all the references according to the requirements of this Journal.  

Author Response

Dear reviewer,

Thank you for these pertinent comments

1) The introduction section was revised. Irrelevant text was deleted L30-37; L42-41; L58-60; L72-L75; L87-89; L95-L100. Partially for better understanding of the article aims L87-L95 was added.

2) The citation format was corrected. We are sorry for that mistake.

3) The used ratios and concentrations were selected according to typical additives use in meat industry. The increasing concentration is based on our previous research. The references were added. L114-L120.

4) The conclusion was corrected. We incorporated major finding of the article. Also, further research is added at the end of conclusion. L429-L448.

5) The references was corrected according to the journal style.

We also performed professional editing to ensure that the text is well phrased and free from typographical and grammatical errors. All changes are highlighted in the manuscript.

Reviewer 3 Report

In the manuscript was described interesting new immunohistiochemical methods for identification of allergenic milk proteins used as additives in different kind of meat products. Compared to well known commercial test used for milk protein detection (Veratox ELISA kit) the elaborated methods were less sensitive and specific. Despite of this, the utilisation for directly visualisation of possible presence allergenic proteins in meat make immunohistochemical methods very useful as screening tool in some research. In particular will be helpful in fast control of immunoreactivity changes of milk additives in meat matrix after each step o processing. From that point of view the manuscript is interesting and should also interest "Food" readers. Nevertheless, some minor revision are necessary:
The methods of calculation sensitivity and specificity should be described with more details since they are not very common and citation, isn't sufficient. 
In this context also the interpretation of results isn't clear and need additional explanation.

Author Response

Dear reviewer,

Thank you for these pertinent comments

The description of the methods of calculation of sensitivity and specificity was added on L344-L347. Also, interpretation was included on L350 and L374-L377 as well as in conclusion L443-L448.

We think that great percentage of false-negative results (specify) is important finding because it shows that casein can be masked by meat products.

Tthe aims and conclusions were rephrased according to all reviewer comments. The results of sensitivity and specificity are also described.

Round 2

Reviewer 2 Report

All the questions were addresed in this vestion.